# A Comparison of Pre-Emptive Co-Amoxiclav, Postoperative Amoxicillin, and Metronidazole for Prevention of Postoperative Complications in Dentoalveolar Surgery: A Randomized Controlled Trial

**DOI:** 10.3390/ijerph19074178

**Published:** 2022-03-31

**Authors:** Giath Gazal, Khalid H. Al-Samadani, Hamdi Mohammed Alsaidalani, Ghofran Ali Karbouji, Abdullah Mohammed Alharbi

**Affiliations:** 1Department of Oral and Maxillofacial Surgery, Taibah University, Al Madinah Al Munawwarah 42353, Saudi Arabia; hmalsaidalani@gmail.com (H.M.A.); ghofrankarbouji@gmail.com (G.A.K.); abdullaharbi@gmail.com (A.M.A.); 2Department of Restorative Dental Science, College of Dentistry, Taibah University, Al Madinah Al Munawwarah 42353, Saudi Arabia; kalsamadani@gmail.com

**Keywords:** surgical dental extraction, dry socket, oral antibiotics

## Abstract

Objective: To compare the effectiveness of different oral antibiotics for prevention of dry socket and infection in adults following the surgical extraction of teeth under LA. Methods: This randomized controlled study was conducted from 10 September 2020 until 10 May 2021. Forty-six patients were randomly allocated to three groups. Sixteen patients were in the postoperative co-amoxiclav (625 mg) group, fifteen in the preoperative co-amoxiclav (625 mg) plus postoperative metronidazole (500 mg) group and fifteen in the preoperative co-amoxiclav (625 mg) plus postoperative amoxicillin (500 mg) group. Evaluation of the postoperative signs of alveolar osteitis and infection was made by a dental surgeon five days postoperatively. Evaluation of the post-surgical extraction pain was made by patients immediately and five days postoperatively on standard 100 mm visual analogue scales (VAS). Furthermore, difficulty of surgery was recorded for all patients immediately postoperatively using (VAS). Results: all antibiotics used in this study were effective. Only 15% of patients had painful alveolar osteitis and 2% had oral infections. There was no significant decrease in the number of patients with severe alveolar osteitis or infection for co-amoxiclav plus metronidazole and co-amoxiclav plus amoxicillin groups compared to co-amoxiclav group at 5 days post-operation (*p*-values: 0.715, 0.819 & 0.309). Clinically, metronidazole was more effective in protecting the extracted tooth socket from alveolar osteitis compared to co-amoxiclav and amoxicillin. Moreover, there were significant decreases in mean pain scores at 5 days post-operation compared with the levels of pain immediately after surgery (*p*-value: 0.001). Conclusions: Administration of a single preoperative dose of co-amoxiclav with a full postoperative dose of amoxicillin or metronidazole was more effective than conventional treatment with postoperative co-amoxilcalv in reducing the incidence of both alveolar osteitis and infection after surgical extractions. However, these differences were not statistically significant. Interestingly, patients in metronidazole group had the lowest incidence of dry socket.

## 1. Introduction

Oral antibiotics are very important after surgical extraction in order to speed up the healing process [1,2]. There are too many factors which can contribute to infection or dry socket after surgical extraction. The most important factor is the formation of necrotizing tissue at the site of surgical extraction [3,4,5,6]. Causes of necrosis include inadequate blood supply (as in infarcted tissue), bacterial infection, traumatic injury and hyperthermia) of the bone. Since surgical extraction requires raising the flap and removing bone on the buccal side using a low- and high-speed drill, this implies trauma to adjacent tissues [1,2,3,4,5,6]. Failure to raise the mucoperiosteal flap well leads to its rupture [7]. It is known that the bone is nourished by blood supply from both bone marrow and periosteum [5,6,7,8]. Therefore, the rupture of the periosteum or its burning during the removal of the bone with the rotating surgical burs results in the formation of foci of periosteal necrosis [6,8]. These necrotic tissues are considered as blood-deprived and dehydrated areas, impeding the process of healthy granulation, and an easy target for bacteria to attack and multiply at the site of the surgical extraction [3,5,8]. Furthermore, surgical extractions are mostly carried out for grossly decayed teeth with apical lesions and abscesses. However, the number of surgical extractions of healthy teeth is less and is limited to the necessities of orthodontic treatments.

Hence the importance of giving antibiotics to patients undergoing surgical extraction under local anesthetic in order to reduce the chance of their exposure to dry socket and infection [1,4,7]. As long as there are odontogenic infections, elevation of the flaps and removal of the bone, there is a possibility of dry socket and involvement of facial spaces. Odontogenic infections are usually mixed with both aerobes and anaerobes (aerobic 25% and anaerobic 75%) [2,5,9].

A study by Yoshida et al. [2] was carried out to examine the effect of preoperative amoxicillin on surgical site infections in Japanese patients who had surgical extraction of lower third molars. Results suggested that amoxicillin beginning 1 h before surgery and lasting for 3 days might be sufficient to prevent surgical site infection (SSI).

However, the findings of the Isiordia-Espinoza et al. [4] study concluded that the routine use of systemic metronidazole to prevent surgical site infection and/or dry socket in healthy patients undergoing third molar surgery is not recommended.

In the lights of these facts, there is still a lack of evidence supporting the use of preoperative oral antibiotics for reducing the occurrence of dry socket and infection following the surgical removal of teeth.

The aim of this study was to compare the efficacy of a preoperative single dose of co-amoxiclav plus postoperative metronidazole/amoxicillin with postoperative co-amoxiclav (conventional therapy) in prevention of dry socket. The null hypothesis was oral administrations of preoperative co-amoxiclav plus postoperative metronidazole/amoxicillin and postoperative co-amoxiclav are equally effective in reducing the occurrence of post-surgical tooth extraction infection and dry socket.

## 2. Materials and Methods

This randomized controlled study was conducted from the 15 October 2020 until the 3 April 2021. The design and performance of this clinical study was done in accordance with the principles of the Declaration of Helsinki. Taibah Dental School Research Ethics Committee had approved the study. The trial registration number is NCT03844776. Written consent had been obtained from 60 patients who attended the Oral and Maxillofacial department. Inclusion criteria for enrolling patients in this study were ASA (American Society of Anesthesiologists, Schaumburg, IL, USA) class I or II patients who were healthy or with mild systemic disease and had no risk from administration of LA with adrenaline; aged 18–70 years; presenting for elective surgical single-tooth extraction. Exclusion criteria involved patients who were sensitive to co-amoxiclav, amoxicillin, or metronidazole; having simple or multiple teeth extractions; and having teeth with reversible pulpitis. Slips of paper were labeled with postoperative 625 mg co-amoxiclav (control group), preoperative 625 mg co-amoxiclav plus postoperative 500 mg amoxicillin, or preoperative 625 mg co-amoxiclav plus postoperative 500 mg metronidazole using computer-generated random number and placed in sequentially numbered envelops. The secretary of the clinic, who was not associated with the study, did this work. When all the screening procedures were completed and the eligibility of the patient was confirmed, the patient was allocated the next numbered envelope. This was opened by a dental assistant not associated with the study and the named antibiotic on the slip of paper and was given to the patient. The slip was placed back into the envelope and put back into the patient’s records. This ensured that both the patients and the investigator were blinded to the study group assignment.

The patients in the study groups were pre-medicated with a preoperative single dose of co-amoxiclav (625 mg) at least 1 h before administering the local anesthetic for surgical extractions. Then, patients in all groups were post-medicated with co-amoxiclav (625 mg), amoxicillin (500 mg) or metronidazole (500 mg) plus 0.2% chlorhexidine mouthwash following the surgery for 5 days (Figure 1). The co-amoxiclav, amoxicillin, and metronidazole doses used for this study were chosen according to manufacturer’s recommendation and set at such a level as to keep the minimum side effects [9,10,11]. Local anesthesia (1.8 mL mepivacaine 2% with epinephrine 1:100,000.) was administered to the patient after sitting on a dental chair. A standardized surgical technique was used for tooth extraction for all patients in the study. The surgical technique included a buccal full-thickness flap, buccal bone removal, tooth extraction, alveolar socket debridement, irrigation with approximately 5 mL of normal saline, and placement of stitches. Each patient was examined for signs and symptoms of alveolar osteitis (dry socket) and infection 5 days post-operation by using a clinical evaluation scale. This scale was developed by the first author (Giath Gazal) basing on the chapter 178, Post-Extraction Pain and Dry Socket (Alveolar Osteitis) Management [12]. This scale was used by a trained and completely independent surgeon for the whole process. Signs of alveolar osteitis included empty-looking socket, bone exposure, and soft tissue inflammation. While the symptoms included throbbing pain, intraoral halitosis, and bad test. The recorded signs of surgical site infection (SSI) were oral swelling, tenderness to touch, drainage of pus, and limitation of mouth opening. The degree of difficulty of surgical extraction was recorded for all patients immediately after the end of surgery. In addition, immediate and 5 days post-operative pain levels were assessed using the visual analog scale (VAS).

### Statistical Analysis

A study with 45 subjects was reported to have 80% power to detect a difference in success rate of 21% in a continuous outcome measure assuming a significance level of 5% and a correlation of 0.5 between responses from the different subjects [13]. So, a total of 60 patients were recruited for this study. Statistical analysis was performed using a software package (SPSS; version 20, SPSS Inc., Chicago, IL, USA). These statistical tests were descriptive analysis tests, one-way ANOVA, and non-parametric tests.

## 3. Results

Of the 60 recruited, 14 patients were excluded by the dental surgeon because they were considered unsuitable for inclusiong in this study (four had upset stomach after administering the preoperative antibiotic doses, three fainted after local anesthetic injection, three had their teeth extracted without the need for surgical intervention four refused extraction after local anesthetic injection). The final sample size included 46 patients. Sixteen took a postoperative dose of 625 mg of co-amoxiclav (control group), fifteen had preoperative dose of 625 mg of co-amoxiclav and postoperative dose of 500 mg of metronidazole, and fifteen took preoperative dose of 625 mg of co-amoxiclav and postoperative dose of 500 mg of amoxicillin (Figure 1). It was considered appropriate to use non-parametric tests because the analyzed data is nominal.

This study included 31 (67.5%) male and 15 (32.5%) female patients. The age range of patients was from 18 to 60 years with a mean age of 37.5 years. There were no significant differences in the distribution of male and female or the ages of patients between the three study groups (*p*-values = 0.699 and 1.000). As for the site of surgery, 80% of patients had surgical extractions of mandibular teeth, while the remaining 20% had surgical removal of maxillary teeth. Statically, there were no differences in the distribution of site of surgeries for the co-amoxiclav, co-amoxiclav plus metronidazole, and co-amoxiclav plus amoxicillin groups (*p*-value = 0.815).

Difficulty of surgery was assessed by using descriptive statistics. Twenty-nine surgical extractions were rated as moderate to very difficult. However, seventeen surgical cases was reported with mild difficulties. Evaluations carried out by an independent observer using VAS. Statistically, there were no significant differences in the number and degree of difficulty of surgical extractions between the study (*p*-values from one-way ANOVA: 0.132, 0.253, & 0.410).

Immediately and 5 days post-operation pain intensity was mild. Mean pain scores were consecutively 30 and 10. There were no statistically significant differences between the mean pain scores for co-amoxiclav, co-amoxiclav plus metronidazole, and co-amoxiclav plus amoxicillin groups immediately and 5 days post-operation (*p*-values from one- way ANOVA: 0.648, 0.141). However, there were statistically significant decreases in mean pain scores at 5 days post-operation compared with the levels of pain immediately after surgery (*p*-values from *t*-test: 0.001). Clinically patients were comfortable 5 days after the surgery.

### 3.1. The Main Results

The three antibiotics regimens used in this study were effective in reducing the number of patients with dry socket (DS) after surgical extraction. The number of patients who did not show signs/symptoms of dry socket (DS) ranged from 33 (72%) to 45 (98%). However, the number of patients with signs/symptoms of dry socket ranged from 1 (2%) to 13 (28%).

Overall, 39 (85%) of the study participants did not experience painful signs or symptoms related to alveolar osteitis (dry socket). However, 7 (15%) of patients had one or more painful signs or symptoms of dry socket.

Chi-Square tests were applied to compare the effectiveness of oral antibiotics for reducing the dry socket signs/symptoms following surgical extraction of teeth in adults under local anaesthetic. Dry socket signs included empty looking-socket, bone exposure, and soft tissue inflammation. There were no significant differences in the number of patients with or without signs of dry socket in the co-amoxiclav, co-amoxiclav plus metronidazole, and co-amoxiclav plus amoxicillin groups (*p*-values = 0.715, 0.819 and 0.309, Table 1).

Notably, there were six (13%) patients in the co-amoxiclav group who had soft tissue inflammation 5 days after surgery compared to four (9%) patients in the co-amoxiclav + amoxicillin group and two (4%) patients in the co-amoxiclav + metronidazole group. However, this difference was not statically significant. Clinically, patients given metronidazole and amoxicillin courses had less severe signs of dry socket than those given co-amoxiclav.

Dry socket symptoms 5 days post-operation, including throbbing pain, intraoral halitosis, and bad test, were analyzed using χ^2^ test. There were no significant differences in the number of throbbing pain, and intraoral halitosis in patients who received either co-amoxiclav, or co-amoxiclav + amoxicillin antibiotics (*p* = 0.171 & 0.073, Table 2). However, the number of patients who had bad test 5 days post-operation was the lowest in the co-amoxiclav + metronidazole group compared to the other treatment groups. This difference was statically significant (*p* = 0.026).

Clinically, a combination of preoperative single dose of co-amoxiclav (625 mg) with postoperative full course of metronidazole (500 mg) showed greater protection of extracted tooth socket from alveolar osteitis compared to full courses of amoxicillin (500 mg) or co-amoxiclave (625 mg) 5 days post-operation (Figure 2).

### 3.2. Surgical Site Infection (SSI)

The total incidence of SSI was recorded and analyzed statistically including oral swelling, tenderness to touch, drainage of pus, and limitation of mouth opening. There were no significant differences between the groups in respect of the infection complications (*p* = 0.997, 0.063, 0.384, and 0.354, Table 3).

In general, the three regimens of antibiotics were effective in reducing the incidence of infection after surgical removal of teeth. There were two patients in each study group with oral swelling, nine patients in both co-amoxiclav and co-amoxiclav plus amoxicillin groups with tenderness to touch, and eight patients with limitation of mouth opening, half of them were in the co-amoxiclav group. However, only one (2%) patient in the co-amoxiclav group had drainage of pus (Table 3, Figure 3).

Practically, patients who took pre-emptive antibiotics in addition to either metronidazole or amoxicillin had less incidences of surgical site infection (SSI) than those given only co-amoxiclav.

## 4. Discussion

Clinically, the findings of this study revealed that all the oral antibiotics used were effective and reduced the incidence and severity of the post-surgical extraction dry socket and infection. A total of 85% of patients who underwent surgical extractions had neither typical signs nor painful symptoms of dry socket. However, 15% of patients had severe dry socket. Remarkably, the incidence of infection associated with the discharge of pus at the site of extraction was 2%. The findings of this study are consistent with the result of a systematic review carried out by Tarakji et al. [5] which showed that the overall frequency of dry socket after teeth extraction was 3.2%. The incidence of dry socket after non-surgical extraction was 1.7%, while it was 15% after surgical extraction. Moreover, the incidence of dry socket was significantly higher in smokers, 12%, than in non-smokers, 4%. Therefore, there is an association between the frequency of smoking cigarettes and the incidence of dry socket. Smoking increases the temperature of the mouth and causes dehydration at the extraction site, which in turn increases the possibility of a blood clot separating from extraction site and exposing the bone socket [1,5,14,15]. The same damage occurs when traumatic surgical extractions perform by using high speeds to remove the bone or separate the roots of the teeth [1,16]. As a consequence, dehydration occurs at the site of the extraction causing weak adhesion of the clot to the walls of the socket. Another possibility of dry socket and infection after surgical extraction is not using proper irrigation with normal saline [4,5,17]. This means that bony and dental debris remain inside the socket of extracted tooth, which prevent the formation of a healthy blood clot [7,9,11]. Clinically, there is a socket filled with a blood clot containing a collection of dirty tissue that forms a fertile focus for the multiplication of aerobic and anaerobic bacteria. This bacterial spot will eat the blood clot and dissolve it to expose the alveolar bone and fill it with infective substances [17,18,19,20]. Bacteria attack on the site of extraction increases dehydration and accelerates the formation of microscopic clots in the bony surfaces of the extracted tooth socket [4,8]. These microscopic clots impede blood flow at the site of extraction, causing a decreased ability to eliminate bacterial virulence [5,7,21]. Thus, the site of extraction becomes fragile and susceptible to various degrees of infection, the least severe of which is alveolar osteitis (dry socket) and the most severe of which is osteomyelitis.

The last possibility of alveolar osteitis (dry socket) is related to the patient himself. After the surgical extraction, there are a number of patients who do not adhere to the post-operative instructions and do mouth rinses and frequent sputum, which causes the clot to be washed out and the surface of the extracted tooth bone is exposed [3,11,22].

After this logical presentation of the causes of dry socket and infection, we can develop a systematic and effective treatment strategy by covering the points mentioned above. First, it is necessary to perform a good irrigation with normal saline after the surgical extraction in order to get rid of the dental, bony and microbial residues [7,20,23]. Secondly, palpating the site of the extraction to detect the sharp bony edges and smooth them with the bone files [24,25,26,27]. Thirdly, the importance of giving the patient a course of antibiotics in order to eliminate the aerobic and anaerobic microbes that may activate in cases of periosteal injuries [1,5,28]. Fourth, emphasizing to the patient the importance of adherence to the postoperative instructions [28,29,30]. By following these steps and adopting them in dental clinics, the chances of developing alveolitis and infection can be reduced to the minimum. The question that arises now is what is the ideal antibiotic that can be prescribed and used in conditions of dry sockets and infection?

Amoxicillin works by inhabiting the cell wall synthesis of bacteria. It has a broad spectrum similar to ampicillin but is better absorbed and achieves higher tissue concentrations [5,11]. It is considered as beta-lactamase sensitive antibiotic. So, it is not effective against staphylococcal organisms/gram negative anaerobic bacteria (this is its weak point).

Clavulanic acid is irreversible inhibitor of β-lactamase enzymes, well absorbed orally, and combined with amoxicillin [1,7,30].

Co-amoxiclav (Augmentin) is a combination of clavulanic acid and amoxicillin [2,10,13]. The addition of clavulanic acid broadens the antibacterial spectrum of amoxicillin to include the bacteria which produce beta lactamase enzyme [1,2,3,4,5,6]. The beta-lactamase enzymes deactivate antibiotics such as penicillin and amoxicillin by hydrolyzing the peptide bond of beta-lactam ring [1,2,3].

Metronidazole is very effective against anaerobic organisms, inhibits protein synthesis by breaking the nuclei’s DNA. Therefore, it causes cell death in susceptible organisms [2,3,4,5,6].

Augmentin (amoxicillin/clavulanate) is the first choice as an antibiotic for most common types of infections. However, it may not work against more serious or uncommon types of infections. Metronidazole (Flagyl) is good at killing a variety of bacteria.

Statistically, this study did not show a significant superiority of any of the antibiotics used to reduce the incidence of alveolitis. Possible reasons for the absence of statistical differences could be the sample size which was too small for detecting any variation or the efficacy of medications was either very similar or too small to make a meaningful change.

Clinically, this study showed that the administration of a single preoperative dose of co-amoxiclav with a full postoperative dose of amoxicillin or metronidazole was more effective than conventional treatment with amoxiclav for reducing the incidence of both dry socket and infection after surgical extractions. Interestingly, patients who took metronidazole had the lowest incidence of dry socket.

The results of this study cast a light on the fact that a single preoperative dose of antibiotic is not sufficient to reduce the occurrence of dry socket and surgical site infection following surgical tooth extraction. Evidence for this finding, a study by Yanine et al. [2] was conducted to investigate the effectiveness of pre-emptive antibiotic for preventing infectious complications following the surgical removal of lower third molars. The outcome of this study revealed that the use of 2 g amoxicillin 1 h before surgery was not effective in reducing the incidence of postoperative infections, when compared to placebo. Furthermore, Khooharo et al. [1] compared the effect of metronidazole and amoxicillin as preoperative single dose treatment with postoperative amoxicillin plus metronidazole (conventional therapy) for prevention of dry socket after surgical removal of lower third molars. The result of their study showed that the single preoperative dose of metronidazole and amoxicillin was not effective in preventing the occurrence of dry socket compared to conventional therapy. 

So, a single preoperative adjunct dose of 625 mg co-amoxiclav (Augmentin) may reduce the incidence of dry socket and infection after surgical extraction, provided it is supplemented with a full postoperative dose of amoxicillin or metronidazole.

There are two possible explanations for recommending the use of metronidazole over both the amoxicillin and co-amoxiclav regimes. First one, the metronidazole used in this study has bone penetration capabilities that are less than the optimal levels for other antibiotics [6]. Also, it has a high level of soft tissue distribution because it contains lipophilic molecules in its chemical formula [7].

This means that metronidazole concentration is high in the superficial bony layer of the extracted tooth socket, periosteum and adjacent soft tissues. This constitutes a strong line of defense against any bacterial attack that may cause necrosis or dissolution of the blood clot.

The second possible account for the superiority of metronidazole over amoxicillin and co-amoxiclav could be as a result of the involvement of anaerobic bacteria in early dissolution of the blood clot formed after surgical extraction, and metronidazole is the unique antibiotic that kills such an infection [1,2,17,29,30].

So, giving a dose of co-amoxiclav before surgery supports the action of both amoxicillin and metronidazole after surgery, in addition to reducing the side effects resulting from the use of a broad-spectrum antibiotic such as co-amoxiclav.

The co-amoxiclav used in this study is bactericidal instead of bacteriostatic. Oral administration of 625 mg co-amoxiclav requires 1.5 h to reach its peak blood concentration [10]. However, it may take up to 48 h of dosing for clinical improvement to appear [1,7,10]. This means that giving a preventive dose of co-amoxiclav will make the extracted tooth socket a healthy and safe area for a blood clot to form. This, in turn, will eliminate any role of aerobic and anaerobic microorganisms in causing dry socket or infection. Furthermore, antibiotics will be useful for elderly patients or those with impaired immune response because they are susceptible to dry socket after dental extraction [8]. However, the mechanical factors that lead to dissolving the formed blood clot such as frequent mouth rinses, smoking, surgical techniques or surgeons with poor experience are still happening and causing a number of postoperative complications in dentoalveolar surgery [27,28,29,30].

Finally, the use of antibiotics might be useful to prevent cases of dry socket after surgical extraction, especially in elderly patients or those with weak immune system. When dry socket occurs, the preferred treatment is to start with saline irrigation and eugenol pastes. Next, application of hyaluronic acid/platelet-rich fibrin and low-level laser therapies which showed a significant reduction in pain and soft-tissue inflammation in the management of dry socket compared to Alveogyl [8,25].

This study has highlighted a new area for further research. This includes investigating if there are analgesic and anti-infective effects of dry socket paste combines topex (20% benzocaine gel), topical clindamycin and surgicel when they are applied at the site of surgery immediately after extraction.

## 5. Conclusions

Overall, 85% of patients had no or mild signs of alveolar osteitis, whilst 15% had severe pain. Only 2% of patients had surgical site infection. This study has shown that the administration of a preoperative single dose of co-amoxiclav plus postoperative amoxicillin or metronidazole was more effective than conventional treatment with postoperative amoxiclav in reducing the incidence of both alveolar osteitis and infection after surgical extractions. However, patients who took metronidazole had the lowest incidence of alveolar osteitis.

**Recommendations:** Pre-emptive antibiotics play an important role in reducing postoperative dry socket and infection associated with surgical tooth extractions under LA. Therefore, it is strongly recommended for all dental surgeons and practitioners in Saudi Arabia to give a single preoperative does of 625 mg co-amoxiclav plus postoperative metronidazole or amoxicillin for their patients who are going to have surgical dental extractions. The number of participants in this study was rather small. Therefore, further study with larger sample size might have more solid results.

## Figures and Tables

**Figure 1 ijerph-19-04178-f001:**
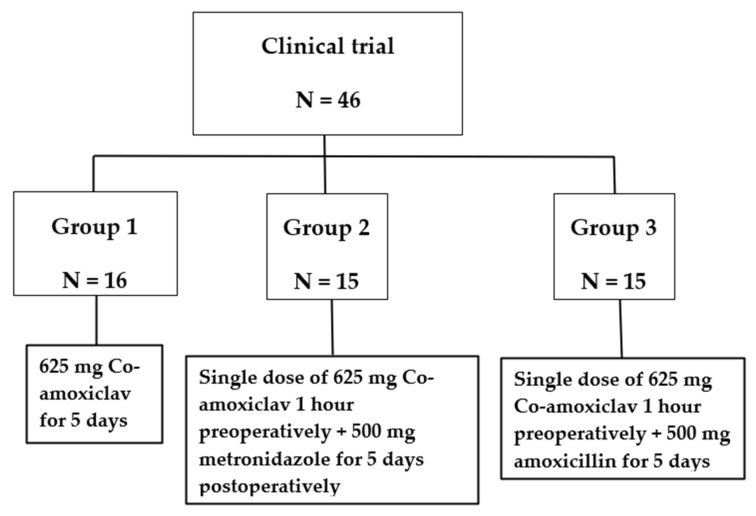
Description of Study design and study groups.

**Figure 2 ijerph-19-04178-f002:**
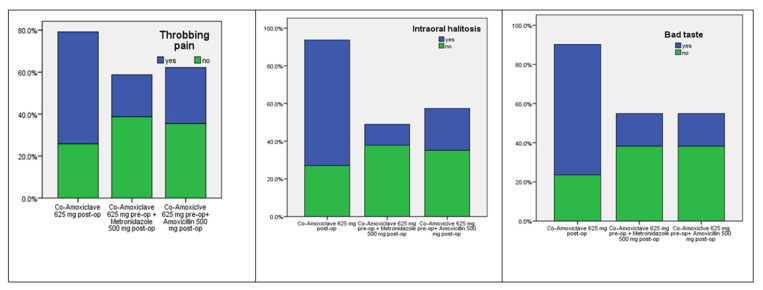
Distribution of the dry socket symptoms occurrence for the patients in the co-amoxiclav, co-amoxiclav plus metronidazole and co-amoxiclav plus amoxicillin groups.

**Figure 3 ijerph-19-04178-f003:**
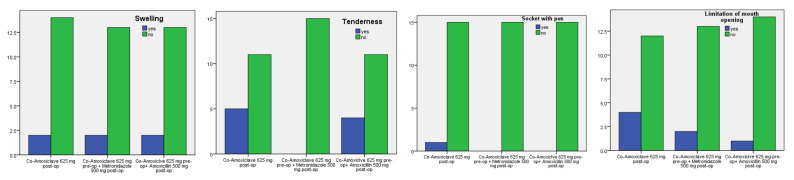
Distribution of surgical site infection (SSI) cases among the oral antibiotics groups.

**Table 1 ijerph-19-04178-t001:** Comparisons of the total incidence of dry socket signs for the patients in the co-amoxiclav, co-amoxiclav plus metronidazole and co-amoxiclav plus amoxicillin groups 5 days post-operation.

		Treatment Groups		Number of Patients % 46 (100%)	X^2^ (DF = 2)	*p*-Value
Dry socket (DS) signs 5 days postoperatively	Empty socket	Co-amoxiclav 625 mg post-op	Yes	4 (9%)	0.672	0.715
			No	12 (26%)		
		Co-amoxiclav 625 mg pre-op + Metronidazole 500 mg post-op	Yes	3 (7%)		
			No	12 (26%)		
		Amoxiclv 625 mg pre-op + Amoxicillin 500 mg post-op	Yes	2 (4%)		
			No	13 (28%)		
	Bone exposure at the site of extraction	Co-amoxiclav 625 mg post-op	Yes	2 (4%)	0.399	0.819
			No	14 (31%)		
		Co-amoxiclav 625 mg pre-op + Metronidazole 500 mg post-op	Yes	3 (7%)		
			No	12 (26%)		
		Amoxiclv 625 mg pre-op + Amoxicillin 500 mg post-op Yes	yes	2 (4%)		
			No	13 (28%)		
	Soft tissue inflammation	Co-amoxiclav 625 mg post-op	Yes	6 (13%)	2.349	0.309
			No	10 (22%)		
		Co-amoxiclav 625 mg pre-op + Metronidazole 500 mg post-op	Yes	2 (4%)		
			No	13 (28%)		
		Amoxiclv 625 mg pre-op + Amoxicillin 500 mg post-op	Yes	4 (9%)		
			No	11 (24%)		

**Table 2 ijerph-19-04178-t002:** Comparisons between different regimens of oral antibiotics and dry socket symptoms occurrence.

		Treatment Groups		Number of Patients % 46 (100%)	X^2^ (DF = 2)	*p*-Value
Dry socket (DS) symptoms 5 days postoperatively	Throbbing pain	Co-amoxiclav 625 mg post-op	Yes	8 (18%)	3.53	0.171
			No	8 (18%)		
		Co-amoxiclav 625 mg pre-op + Metronidazole 500 mg post-op	Yes	3 (7%)		
			No	12 (26%)		
		Co-amoxiclav 625 mg pre-op + Amoxicillin 500 mg post-op	Yes	4 (9%)		
			No	11 (24%)		
	Intraoral halitosis	Co-amoxiclav 625 mg post-op	Yes	6 (13%)	5.23	0.073
			No	10 (22%)		
		Co-amoxiclav 625 mg pre-op + Metronidazole 500 mg post-op	Yes	1 (2%)		
			No	14 (31%)		
		Co-amoxiclav 625 mg pre-op + Amoxicillin 500 mg post-op Yes	yes	2 (4%)		
			No	13 (28%)		
	Bad test	Co-amoxiclav 625 mg post-op	Yes	8 (18%)	7.28	0.026
			No	8 (18%)		
		Co-amoxiclav 625 mg pre-op + Metronidazole 500 mg post-op	Yes	2 (4%)		
			No	13 (28%)		
		Co-amoxiclav 625 mg pre-op + Amoxicillin 500 mg post-op	Yes	2 (4%)		
			No	13 (28%)		

**Table 3 ijerph-19-04178-t003:** Relationship between different regimens of oral antibiotics and SSI occurrence.

		Treatment Groups		Number of Patients % 46 (100%)	X^2^ (DF = 2)	*p*-Value
Surgical site infection (SSI) 5 days postoperatively	Oral swelling	Co-amoxiclav 625 mg post-op	Yes	2 (4%)	0.006	0.997
			No	14 (32%)		
		Co-amoxiclav 625 mg pre-op + Metronidazole 500 mg post-op	Yes	2 (4%)		
			No	13 (28%)		
		Co-amoxiclav 625 mg pre-op + Amoxicillin 500 mg post-op	Yes	2 (4%)		
			No	13 (28%)		
	Tenderness to touch	Co-amoxiclav 625 mg post-op	Yes	5 (11%)	5.517	0.063
			No	11 (23.5%)		
		Co-amoxiclav 625 mg pre-op + Metronidazole 500 mg post-op	Yes	0 (0%)		
			No	15 (33%)		
		Amoxiclv 625 mg pre-op + Amoxicillin 500 mg post-op	yes	4 (9%)		
			No	11 (23.5%)		
	Drainage of pus	Co-amoxiclav 625 mg post-op	Yes	1 (2%)	1.917	0.384
			No	15 (32.7%)		
		Co-amoxiclav 625 mg pre-op + Metronidazole 500 mg post-op	Yes	0 (0%)		
			No	15 (32.7%)		
		Amoxiclv 625 mg pre-op + Amoxicillin 500 mg post-op	Yes	0 (0%)		
			No	15 (32.7%)		
	Limitation of mouth opening	Co-amoxiclav 625 mg post-op	Yes	4(9%)	2.078	0.354
			No	12(26%)		
		Co-amoxiclav 625 mg pre-op + Metronidazole 500 mg post-op	Yes	2(4%)		
			No	13(28%)		
		Amoxiclv 625 mg pre-op + Amoxicillin 500 mg post-op	Yes	1(2%)		
			No	14(31%)		

## Data Availability

The data that support findings of this study are available on request from the corresponding author.

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
