# Peer review of "A Comparison of Pre-Emptive Co-Amoxiclav, Postoperative Amoxicillin, and Metronidazole for Prevention of Postoperative Complications in Dentoalveolar Surgery: A Randomized Controlled Trial"

_ijerph, 2022, doi:10.3390/ijerph19074178_

Round 1

Reviewer 1 Report

I for one do not subscribe to the premise of administering antibiotics, either preoperatively or postoperatively, to healthy dental surgery patients, unless they show signs of infection (and not even dry socket). We, the doctors, should avoid over-using antibiotics and prevent the bacterial resistance they cause. If we care so much about the reduction of post-surgical complications, there are many good and effective alternatives, such as chlorhexidine mouthwash, or better than that chlorhexidine gel, or perhaps other antiseptic mouthwashes that can reduce the risk of post-surgical infection or dry socket.

Apart from the poor premise, the study had major methodological shortcomings:

The sample size was not based on any a priori calculations. There was no sample size calculation.

The sample was quite small.

There was no control group (i.e., groups without antibiotics, with or without other antiseptic materials).

Many details were missing. Please revise the paper in accordance with the CONSORT checklist. Apply all the CONSORT items and all its sub-items to the paper.

Author Response

Reviewer 1:
1- Comment NO 1:The sample size was not based on any a priori calculations. There was no sample size calculation.

Answer:

There were sample size calculation based on the study below:

  1. Bortoluzzi MC1, Capella DL, Barbieri T, Pagliarini M, Cavalieri T, Manfro R (2013) A single dose of amoxicillin and dexamethasone for prevention of postoperative complications in third molar surgery: a randomized, double-blind, placebo controlled clinical trial. J Clin Med Res. 2013 Feb;5(1):26-33.

However, the number of participants who were excluded from this study was high due to non-compliance with inclusion criteria. For that I add in the recommendation section the following sentence (The number of participants in this study was rather small. Therefore, further study with larger sample size might have more solid results.).

Study’s power was reduced to 80% to match our sample size

2- Comment NO 2: There was no control group

Answer:

This study included the Control Group, which is the participants who were given Augmentin postoperatively. Because the gold standard applied globally is that any surgical procedure that includes removing part of the alveolar bone must give the patient an antibiotic. Please see the figure 1

3- Please revise the paper in accordance with the CONSORT checklist. Apply all the CONSORT items and all its sub-items to the paper.

Answer:

Yes I did revise the paper with the CONSORT checklist. Please see figure 1. This study was approved by ClinicalTrials.gov. Trial registration number: NCT03844776.

Reviewer 2 Report

The manuscript on A comparison of pre-emptive Co-amoxiclav, postoperative Amoxicillin, and Metronidazole for prevention of postopera- tive complications in dentoalveolar surgery: a randomized con- trolled trial is a clinically relevant study and have conducted and written reasonably well. I have a few concerns regarding the manuscript 

  1. With respect to postoperative complications, many confounding factors are there such as smoking and patient compliance etc. In the methodology there was no mention about whether smokers are included or not, how did you ensure the patient compliance regarding medication and othe post operative instructions, the difficulty level of extraction etc. These data are also important in interpreting the results.
  2. In 2.1 section regarding statistical test used authors have mentioned one of the tests used was one way anova, but in results tables no where I could find any evidence for this test.
  3. In the result section 3.1 first paragraph authors have written that “The number of patients who did not show signs/symptoms of dry socket (DS) ranged from 33 to 45, with mean number of 39 ±6 (M±SD). However, the number of patients with signs/symptoms of dry socket ranged from 1 to 13 with mean number of 7±6 (M±SD)”. Expressing a nominal data in mean and SD is not logical instead it can be expressed in percentage.
  4. In the methodology authets have mentioned that  VAS score was used for assessing the postoperative pain assessment, but in the results all this was not given instead all the comparisons were done in nominal way( yes/no) even for post op pain.
  5. Dry socket symptom, bad taste is written as bad test both in the methodology and result table.
  6. In the results, the statisticaly significant findings was observed only for the parameter ‘ bad taste’ where the bad taste was reported significantly less  in both the groups where preoperative medications was given as compared to the first group where No Pre op medications were give given and there was no difference in percentage between the groups where post op medications was metronidazole or amoxicillin. But the authers interpretation of the results “ Clinically, a combination of preoperative single dose of co-amoxiclav (625mg) with postoperative full course of metronidazole (500mg) showed greater protection of ex- tracted tooth socket from alveolar alveolar osteitis compared to full courses of amoxicillin (500 mg) or co-amoxiclave (625 mg) 5 days postoperatively (figure 2)” is not agreeing with the actual results.According to the data No superiority can be claimed for metronidazole group over amoxicillin group.
  7. Figure three legend can be more explanatory 
  8. In the end of discussion authors have written that “This study has highlighted a new area for further research. This include investigating if there are analgesic and anti-infective effects of dry socket paste with topex and surgical when they are applied at the site of surgery immediately after extraction”. This statement is not agreeing with study results as the study didn’t evaluate any topical medications.

Author Response

Reviewer 2:

Comment NO 1: 1.With respect to postoperative complications, many confounding factors are there such as smoking and patient compliance etc. In the methodology there was no mention about whether smokers are included or not, how did you ensure the patient compliance regarding medication and the postoperative instructions, the difficulty level of extraction etc. These data are also important in interpreting the results

Answer:

For smoking, we did not consider it as an exclusion factor from the study while postoperative instructions were delivered to all patients in the form of a leaflet. The difficulty of the surgical procedure has been evaluated in the results section. See below: result page:4

Difficulty of surgery was assessed by using descriptive analysis test. Twenty-nine surgical extractions were rated as moderate to very difficult. However, seventeen surgical cases was reported with mild difficulties. Evaluations carried out by an independent observer using VAS.

Comment NO 2: In 2.1 section regarding statistical test used authors have mentioned one of the tests used was one way anova, but in results tables no where I could find any evidence for this test.

 Answer:

Postoperative pain assessment was evaluated in two phases, the first immediately after surgery and the second five days after surgery. This was done for the three study groups, and since there were no significant statistical differences, the results were presented in writing and not placed in a table in order to avoid prolongation.

Comment NO 3: In the result section 3.1 first paragraph authors have written that “The number of patients who did not show signs/symptoms of dry socket (DS) ranged from 33 to 45, with mean number of 39 ±6 (M±SD). However, the number of patients with signs/symptoms of dry socket ranged from 1 to 13 with mean number of 7±6 (M±SD)”. Expressing a nominal data in mean and SD is not logical instead it can be expressed in percentage

Answer:

I did correct it. See below

The number of patients who did not show signs/symptoms of dry socket (DS) ranged from 33(72%) to 45 (98%). However, the number of patients with signs/symptoms of dry socket ranged from 1 (2%) to 13(28%).

Comment NO 4: In the methodology authors have mentioned that VAS score was used for assessing the postoperative pain assessment, but in the results all this was not given instead all the comparisons were done in nominal way( yes/no) even for post op pain.

Answer:

I do agree with the reviewer but the only reason why we did not put such a result in a table was: There were no significant statistical differences in addition to avoiding prolongation.

Comment NO 5 & 6

Dry socket symptom, bad taste is written as bad test both in the methodology and result table. In the results, the statisticaly significant findings was observed only for the parameter ‘ bad taste’ where the bad taste was reported significantly less in both the groups where preoperative medications was given as compared to the first group where No Pre op medications were give given and there was no difference in percentage between the groups where post op medications was metronidazole or amoxicillin. But the authors interpretation of the results “ Clinically, a combination of preoperative single dose of co-amoxiclav (625mg) with postoperative full course of metronidazole (500mg) showed greater protection of extracted tooth socket from alveolar alveolar osteitis compared to full courses of amoxicillin (500 mg) or co-amoxiclave (625 mg) 5 days postoperatively (figure 2)” is not agreeing with the actual results. According to the data No superiority can be claimed for metronidazole group over amoxicillin group.

Answer:

It was indicated in the conclusion that both amoxicillin and metronidazole are superior to co-amoxiclave clinically and not statistically. This means that the number of patients who had severe signs and symptoms of dry socket was greater in the co-amoxiclave than in the amoxicillin and metronidazole groups. This was extracted from the tables.

Comment NO 7:

Figure three legend can be more explanatory

Answer:

The figure 3 cannot be made explanatory further because the sample size is not large.

Comment NO 8:

In the end of discussion authors have written that “This study has highlighted a new area for further research. This include investigating if there are analgesic and anti-infective effects of dry socket paste with topex and surgical when they are applied at the site of surgery immediately after extraction”. This statement is not agreeing with study results as the study didn’t evaluate any topical medications

Answer:

Our study was focused on antibiotics. This is true, but what we mentioned is a new research idea that can attract new researchers.

Reviewer 3 Report

Dear authors,

I have read with interest your work entitled "A comparison of pre-emptive Co-amoxiclav, postoperative Amoxicillin, and Metronidazole for prevention of postoperative complications in dentoalveolar surgery: a randomized controlled trial".

In my opinion, the experiments were well designed and conducted. The methods used are appropriate to answer the research question posed. However, the report contains inaccuracies and errors that must be corrected. Below you will find questions and suggestions for changes/additions regarding the reviewed work. 

A.d. 2.1. Statistical analysis

Calculate the statistical power for your research. Citing the calculation for previously published works (Ref, 13) is not sufficient. 

Rename "descriptive analysis test" to "descriptive statistics". Values such as percentage, mean, or SD are not derived from statistical testing.  

Ad. 3. Results

You reported that the difficulty of the surgery was assessed. Have you tested the difference in the difficulty of the surgery between the three analysed antibiotics regiments (post co-amoxiclav, pre co-amoxiclav + post metronidazol, pre co-amoxiclav + post amoxixilin)? Was there a difference between these three groups in the surgical techniques used?

You also stated the significant decrease in mean pain score on the 5th day after surgery in comparison to the pain score measured immediately after the intervention. Have you checked if there was a difference in the pain reduction between these groups? Did any of the three types of antibiotic therapy cause the pain to decrease more?

It can be assumed that the severity of the procedure and the type of techniques used have an impact on the amount of pain experienced. Moreover, it can influence the occurrence of signs or symptoms of DS and SSI that you have studied. This may also affect the results obtained in further analyzes if the three analyzed groups are unequal in terms of the severity of the procedure and thus the perceived pain.

A.d. 3.1. The main results

The three antibiotics regimens used in this study were effective in reducing the number of patients with dry socket (DS) after surgical extraction”. This conclusion is not allowed, because the difference in the incidence of DS in the three investigated groups was not compared with the incidence of the DS in the group of patients with no antibiotics treatment or other treatment which can be used as a control.

The number of patients who did not show signs/symptoms of dry socket (DS) ranged from 33 to 45, with mean number of 39 ±6 (M±SD). However, the number of patients with signs/symptoms of dry socket ranged from 1 to 13 with mean number of 7±6 (M±SD).” It is not clear. Why did you refer to the range of numbers of patients? Below, it can be read that “Overall, 39 (85%) of the study participants did not experience painful signs or symptoms related to alveolar osteitis (dry socket). However, 7 (15%) of patients had one or more painful signs or symptoms of dry socket”. This paragraph should be rewritten for clarity.

I suggest joining Tables 1 and 2 and changing the orientation of the columns and rows. It will be beneficial to order the treatment groups as columns and leave the signs and symptoms in rows. It will increase the clarity of the column and make the comparison of numbers and % easier.

In Figure 2, the percentage instead of counts on the Y-axis should be presented since the numbers of patients in the compared three groups are not equal. 

"Clinically, patients given metronidazole and amoxicillin courses had less severe signs of dry socket than those given co-amoxiclav". What did you mean? Are some sings are less or more severe than others?

"However, the number of patients who had bad test 5 days postoperatively was the lowest in the co-amoxiclav + metronidazole group compared to the other treatment groups. This difference was statically significant (P = 0.026)." This conclusion is not allowed. The Chi-squared test indicated that there was a significant difference in the bad test incidence between three compared groups of different antibiotics regiments, generally. It does not indicate that metronidazol regimen is connected with a lower incidence of the bad test than other regimens investigated. This comment is also valid for the next sentence "Clinically, a combination of preoperative single dose of co-amoxiclav (625mg) with postoperative full course of metronidazole (500mg) showed greater protection of extracted tooth socket from alveolar alveolar osteitis compared to full courses of amoxicillin (500 mg) or co-amoxiclave (625 mg) 5 days postoperatively (figure 2)”.

A.d. 3.3.1. Surgical site infection (SSI)

"In general, the three regimens of antibiotics were effective in reducing the incidence of infection after surgical removal of teeth." This conclusion is not allowed, because the difference in the incidence of infection in the three investigated groups was not compared with the incidence of the infection in the group of patients with no antibiotics treatment or other treatment which can be used as a control.

Figure 3 is difficult to understand. Why did you present median numbers instead of percentages for the nominal variables as symptoms of infection? The type of chart should be changed.

A.d. 4. Disscussion

Discussion should be shorten. A lots of basic information about the subject of the investigation (about the first half of the Discussion section) could be presented in the Introducion. The meaning of the results must be thoroughly discussed. 

Author Response

Reviewer 3:

Comment1: Statistical analysis

Calculate the statistical power for your research. Citing the calculation for previously published works (Ref, 13) is not sufficient.

Answer:

There were sample size calculation based on the study below:

  1. Bortoluzzi MC1, Capella DL, Barbieri T, Pagliarini M, Cavalieri T, Manfro R (2013) A single dose of amoxicillin and dexamethasone for prevention of postoperative complications in third molar surgery: a randomized, double-blind, placebo controlled clinical trial. J Clin Med Res. 2013 Feb;5(1):26-33.

However, the number of participants who were excluded from this study was high due to non-compliance with inclusion criteria. For that I add in the recommendation section the following sentence (The number of participants in this study was rather small. Therefore, further study with larger sample size might have more solid results.).

Study’s power was reduced to 80% to match our sample size

Rename "descriptive analysis test" to "descriptive statistics". Values such as percentage, mean, or SD are not derived from statistical testing

Changes were done. MEAN and SD were deleted from the text

Comment 2: Results

  1. You reported that the difficulty of the surgery was assessed. Have you tested the difference in the difficulty of the surgery between the three analysed antibiotics regiments (post co-amoxiclav, pre co-amoxiclav + post metronidazol, pre co-amoxiclav + post amoxixilin)? Was there a difference between these three groups in the surgical techniques used?
  2. Answer:

Done

Difficulty of surgery was assessed by using descriptive statistics. Twenty-nine surgical extractions were rated as moderate to very difficult. However, seventeen surgical cases was reported with mild difficulties. Evaluations carried out by an independent observer using VAS. Statistically, There were no significant differences in the number and degree of difficulty of surgical extractions between the study ((p-values from one- way ANOVA: 0.132, 0.253, & 0.410).

  1. You also stated the significant decrease in mean pain score on the 5th day after surgery in comparison to the pain score measured immediately after the intervention. Have you checked if there was a difference in the pain reduction between these groups? Did any of the three types of antibiotic therapy cause the pain to decrease more?

Answer:

There was a reduction in pain intensity for all study groups compared to the pain reported immediately after surgery. The reason is that the antibiotics reduced the chance of infection and bone exposure, and as a result, the chance of pain.

  1. It can be assumed that the severity of the procedure and the type of techniques used have an impact on the amount of pain experienced. Moreover, it can influence the occurrence of signs or symptoms of DS and SSI that you have studied. This may also affect the results obtained in further analyzes if the three analyzed groups are unequal in terms of the severity of the procedure and thus the perceived pain.            

Answer:

Because the surgeon who performed the surgical extractions was the same person and the equally difficult cases between groups could justify the absence of peripheral effects that caused biased results

.Comment 3: The main results

  1. “The three antibiotics regimens used in this study were effective in reducing the number of patients with dry socket (DS) after surgical extraction”. This conclusion is not allowed, because the difference in the incidence of DS in the three investigated groups was not compared with the incidence of the DS in the group of patients with no antibiotics treatment or other treatment which can be used as a control

Answer:

Please see the discussion section lines number 6-9

The findings of this study are consistent with the result of a systematic review carried out by Tarakji et al [5]. which showed that the overall frequency of dry socket after teeth extraction was 3.2%. The incidence of dry socket after non-surgical extraction was 1.7%, while it was 15% after surgical extraction.

  1. “The number of patients who did not show signs/symptoms of dry socket (DS) ranged from 33 to 45, with mean number of 39 ±6 (M±SD). However, the number of patients with signs/symptoms of dry socket ranged from 1 to 13 with mean number of 7±6 (M±SD).” It is not clear. Why did you refer to the range of numbers of patients? Below, it can be read that “Overall, 39 (85%) of the study participants did not experience painful signs or symptoms related to alveolar osteitis (dry socket). However, 7 (15%) of patients had one or more painful signs or symptoms of dry socket”. This paragraph should be rewritten for clarity. It was done. It was also recommend it by another reviewer see below:The number of patients who did not show signs/symptoms of dry socket (DS) ranged from 33(72%) to 45 (98%). However, the number of patients with signs/symptoms of dry socket ranged from 1 (2%) to 13(28%).
  2.  
  3.  
  4. Answer:
  5.  
  6. I suggest joining Tables 1 and 2 and changing the orientation of the columns and rows. It will be beneficial to order the treatment groups as columns and leave the signs and symptoms in rows. It will increase the clarity of the column and make the comparison of numbers and % easierAnswer: 
  7. I appreciate your comment, but there is a technical difficulty due to the large number of numbers, especially this article targets practitioners and dental students, and the merging process may lead to difficulty in understanding for the readers
  8.  
  9. In Figure 2, the percentage instead of counts on the Y-axis should be presented since the numbers of patients in the compared three groups are not equal.Sorry, the percentage was mentioned in the text, and the sample is small, and the statistical test did not allow me to display it except in numbers. In any case, the information is clear and sufficient
  10.  
  11. Answer:
  12. "Clinically, patients given metronidazole and amoxicillin courses had less severe signs of dry socket than those given co-amoxiclav". What did you mean? Are some sings are less or more severe than others?

Answer:

This means that the number of patients who had severe signs and symptoms of dry socket was greater in the co-amoxiclave than in the amoxicillin and metronidazole groups. This was extracted from the tables.

  1. "However, the number of patients who had bad test 5 days postoperatively was the lowest in the co-amoxiclav + metronidazole group compared to the other treatment groups. This difference was statically significant (P = 0.026)." This conclusion is not allowed. The Chi-squared test indicated that there was a significant difference in the bad test incidence between three compared groups of different antibiotics regiments, generally. It does not indicate that metronidazol regimen is connected with a lower incidence of the bad test than other regimens investigated. This comment is also valid for the next sentence "Clinically, a combination of preoperative single dose of co-amoxiclav (625mg) with postoperative full course of metronidazole (500mg) showed greater protection of extracted tooth socket from alveolar alveolar osteitis compared to full courses of amoxicillin (500 mg) or co-amoxiclave (625 mg) 5 days postoperatively (figure 2)”.Answer:  
  2. Comment 4: Surgical site infection (SSI)
  3. It was indicated that both amoxicillin and metronidazole are superior to co-amoxiclave clinically and not statistically. This means that the number of patients who had severe signs and symptoms of dry socket was greater in the co-amoxiclave than in the amoxicillin and metronidazole groups. This was extracted from the tables.
  4.  
  1. "In general, the three regimens of antibiotics were effective in reducing the incidence of infection after surgical removal of teeth." This conclusion is not allowed, because the difference in the incidence of infection in the three investigated groups was not compared with the incidence of the infection in the group of patients with no antibiotics treatment or other treatment which can be used as a control. 
  2.  
  3. Answer:
  4. Figure 3 is difficult to understand. Why did you present median numbers instead of percentages for the nominal variables as symptoms of infection? The type of chart should be changedThe figure 3 cannot be made explanatory further because the sample size is not large and there is no other alternative.According to literatures the golden standard for dentoalveolar surgery to give a course of amoxicillin and metronidazole or co-amoxiclav postoperatively. So, in our study we considered the patients who received postoperatively co-amoxiclav as a control group. It is impossible to use a group with no antibiotics because this action is against the hospitals guidelines all over the world.Comment 5: 4. Disscussion
  5.  
  6.  
  7.  
  8. Answer:
  1. Discussion should be shorten. A lots of basic information about the subject of the investigation (about the first half of the Discussion section) could be presented in the Introducion. The meaning of the results must be thoroughly discussed.Answer:
  2. I appreciate this comment but the reason I added basic knowledge because our target are the under& postgraduate dental students in addition to GDPs, junior and seniors oral surgeons. Moreover, I tried to explain thoroughly the mechanism of action of the 3 antibiotics and their role in reducing the incidence of infection and dry socket.
  3.  

Reviewer 4 Report

Dear Authors,

I respect and agree with the intent of this innovative and courageous trial of perioperative medication changes for tooth extraction.

For procedures such as tooth extraction, where there are differences in the degree and site of invasion, background factors can have a considerable impact. The lack of clarification or comparison of background factors (surgical site, gender, age, etc.) between groups can be an alarming issue.

In addition, power is an important issue in conducting RCTs on new drug therapies. As you know, the lack of a statistically significant difference between groups does not mean that a change in medication regimen is not meaningful. However, it may have a certain significance when it comes to reporting that the results are not significantly different.

In this paper, we designed the tolerance for alpha error (= significance level) to be 5 % and for beta error to be 10 % (= power to be 90 %), so the sample size would need to be ‘n = 50’ for each group.

Therefore, you must add the consideration of two possibilities when there is no statistically significant difference: (1) the difference could not be detected because the sample was too small (= beta error), or (2) the effect is too small or not at all to make a significant difference (= the change is not really meaningful).

Int. J. Environ. Res. Public Health is a high quality journal, so we would like to avoid this kind of consideration. It may take some time, but we hope that you will try again with a larger sample size.

As a small detail, there is a place in Table 2 where the word "Yes" is in the wrong place. Please fix it.

Good luck.

Author Response

Reviewer 4:

Comment 1: For procedures such as tooth extraction, where there are differences in the degree and site of invasion, background factors can have a considerable impact. The lack of clarification or comparison of background factors (surgical site, gender, age, etc.) between groups can be an alarming issue.

Answer:

I did add the flowing paragraph to the result section see below:

This study included 31 (67.5%) male and 15 (32.5%) female. The age range of patients was from 18 to 60 years with mean age of 37.5 years. .There were no significant differences in the distribution of male and female or the ages of patients between the three study groups ((p- values =.0.699 &1.000). As for the site of surgery, 80% of patients had surgical extractions of mandibular teeth, while the remaining 20% had surgical removal of maxillary teeth. Statically, there were no differences in the distribution of site of surgeries for the co-amoxiclav, co-amoxiclav plus metronidazole, and co-amoxiclav plus amoxicillin groups (P-value= 0.815).

Comment 2: In addition, power is an important issue in conducting RCTs on new drug therapies. As you know, the lack of a statistically significant difference between groups does not mean that a change in medication regimen is not meaningful. However, it may have a certain significance when it comes to reporting that the results are not significantly different.

In this paper, we designed the tolerance for alpha error (= significance level) to be 5 % and for beta error to be 10 % (= power to be 90 %), so the sample size would need to be ‘n = 50’ for each group.

Therefore, you must add the consideration of two possibilities when there is no statistically significant difference: (1) the difference could not be detected because the sample was too small (= beta error), or (2) the effect is too small or not at all to make a significant difference (= the change is not really meaningful).

Int. J. Environ. Res. Public Health is a high quality journal, so we would like to avoid this kind of consideration. It may take some time, but we hope that you will try again with a larger sample size.

Answer:

I did add the flowing paragraph to the discussionsection see below:

Statistically, this study did not show a significant superiority of any of the antibiotics used to reduce the incidence of alveolitis. Possible reasons for the absence of statistical differences could be the sample size which was too small for detecting any variation or the efficacy of medications was either very similar or too small to make a meaningful change.

As a small detail, there is a place in Table 2 where the word "Yes" is in the wrong place. Please fix it.

Done

Reviewer 5 Report

This paper is well written. The methodology is also appropriate. Thank you very much.

Author Response

There were no comments

many thanks

Round 2

Reviewer 3 Report

Many errors/commnets made in the first round reviews were not sufficiently answered and the manuscript was not sufficiently corrected. 

Author Response

ALL figures were replaced and corrected

Reviewer 4 Report

Dear Authors,

The necessary action has been taken. Once the overlapping text boxes for each group in Figure 1 are corrected, we will have no choice but to publish them.

Thank you.

Author Response

All figures were replaced and corrected
